# A Type of Ferrocene-Based Derivative FE-1 COF Material for Glycopeptide and Phosphopeptide Selective Enrichment

**DOI:** 10.3390/jfb15070185

**Published:** 2024-07-04

**Authors:** Yu Wu, Sen Xu, Fengjuan Ding, Weibing Zhang, Haiyan Liu

**Affiliations:** 1School of Chemistry & Molecular Engineering, East China University of Science and Technology, Shanghai 200237, China; y30190255@mail.ecust.edu.cn (Y.W.); y10180216@mail.ecust.edu.cn (S.X.); 2Shanghai Key Laboratory of Functional Materials Chemistry, School of Chempistry & Molecular Engineering, East China University of Science and Technology, Shanghai 200237, China

**Keywords:** covalent organic framework materials, simultaneously enrichment, phosphopeptides, glycopeptides

## Abstract

In this work, a new type of FE-1 COF material is prepared by a reversible imine condensation reaction with diaminoferrocene and diaminodiformaldehyde as materials. The material is connected by imine bonds to form a COF skeleton, and the presence of plenty of nitrogen-containing groups gives the material good hydrophilicity; the presence of metal Fe ions provides the material application potential in the enrichment of phosphopeptides. According to the different binding abilities of N-glycopeptide and phosphopeptide on FE-1 COF, it can simultaneously enrich N-glycopeptide and phosphopeptide through different elution conditions to realize its controllable and selective enrichment. Using the above characteristics, 18 phosphopeptides were detected from α-casein hydrolysate, 8 phosphopeptides were detected from β-casein hydrolysate and 21 glycopeptides were detected from IgG hydrolysate. Finally, the gradual elution strategy was used; 16 phosphopeptides and 19 glycopeptides were detected from the α-casein hydrolysate and IgG hydrolysate. The corresponding glycopeptides and phosphopeptides were identified from the human serum. It proves that the FE-1 COF material has a good enrichment effect on phosphopeptides and glycopeptides.

## 1. Introduction

Protein is the most important biological organism and the material basis for many important functions and biological processes. The post-translational modification of proteins (PTMs) refers to the process of the chemical modification of expressed protein precursors and side-chain processing by organisms to make them become mature proteins with functional activity. In order to better understand the mechanism of disease and make an early diagnosis, it is necessary for us to conduct a comprehensive analysis of different types of post-translational modifications. The post-translational modifications of proteins mainly include protein phosphorylation and glycosylation [1,2]. Protein phosphorylation exists widely in various cellular activities, including growth, proliferation, differentiation, cytometaplasia and apoptosis [3,4]; protein glycosylation also plays an important role in organisms, participating in specific immunity, signal transduction, protein translation regulation and degradation, cell wall synthesis and other biological processes [5,6,7]. Abnormal protein glycosylation and phosphorylation are closely related to many human diseases, and there are biomarkers with clinical diagnostic significance [8,9,10]. The simultaneous detection and accurate analysis of glycosylation and phosphorylation is of great significance for the diagnosis and treatment of diseases and the discovery of disease biomarkers [11,12,13].

At present, biological mass spectrometry is an important tool for proteomics research. However, due to the complex composition of biological samples, the abundance of glycopeptides and phosphopeptides is very low, and there are many interfering substances, such as high-abundance proteins, surfactants, salt and other impurities [14]. Phosphopeptide or glycopeptide information cannot be obtained directly using MS analysis [15,16]. Therefore, it is very important to perform sample pre-enrichment before mass spectrometry analysis. After protease cleavage, due to the low abundance of phosphorylated peptides, its own electronegative and the interference of a large number of non-phosphorylated peptides, phosphorylated proteins/peptides cannot be effectively identified and analyzed in positive ion modewithout pre-separation. Therefore, it is necessary to use highly selective separation and enrichment techniques for phosphorylated proteins/peptides before mass spectrometry analysis. The entire enrichment process can be explained by the Lewis acid–base theory that suggests carrying out enrichment at a low pH and elution at a high pH to achieve highly selective capture of phosphorylated peptides. At present, the enrichment methods of phosphorylated peptides mainly include metal ion affinity chromatography (IMAC) [17,18], metal oxide affinity chromatography (MOAC) [19,20], ion exchange method (IEC) [21,22,23] and immunoaffinity chromatography [24,25]. The enrichment methods of glycopeptides mainly include lectin affinity chromatography [26,27,28], hydrophilic interaction chromatography [29,30], boric acid affinity chromatography [31] and hydrazide chemistry [32]. Hydrophilic interaction chromatography (HILIC) has the advantages of mild enrichment conditions, high enrichment efficiency, strong selectivity, wide sugar group adaptability and good MS compatibility [33], which has been widely used in the enrichment of glycopeptides.

Ferrocene is an organic transition metal compound with high thermal stability, radiation resistance and aromatic properties. It is an orange-yellow powder at room temperature and smells of camphor. It not only has a unique sandwich structure and aromaticity, but its two cyclopentadienes can also rotate freely [34]. Many derivatives can be synthesized through the electrophilic substitution reactions of ferrocene. Especially in recent years, the successful synthesis of ferrocene-based polymers with high molecular weight has made the synthesis [35,36], performance and application of ferrocene-based polymers research make great progress [37], mainly including polycondensation, graft copolymerization, ring-opening polymerization and other methods [38]. The resulting products have been widely used in electrochemistry [39,40,41], catalysts [42,43,44], nanomaterials [45] and biomaterials preparation [46,47,48]. In 2015, Liu synthesized a new type of ferrocene-based nanoporous organic polymer (FNOPs-1) and found it had good physical, chemical and thermal stability [49]. Guo proposed a method to introduce β-cyclodextrin (β-CD) and ferrocene into the main chain of polyacrylic acid (PAA) and synthesized Fc-PAA-β-CD. The polymer can further self-assemble into nanoparticles in water [50]. But so far, no ferrocene derivatives have been used for the enrichment of biological samples, and related applications are still at a blank stage. HILIC is considered to be the first choice for glycopeptide enrichment because of its high selectivity, high efficiency and good versatility. At the same time, Zr [51], Ti [52], Fe [53] and other metals have been confirmed to have the ability to effectively enrich phosphopeptides in the IMAC technology. Therefore, the combination of HILIC and IMAC may be an effective method to simultaneously enrich glycopeptides and phosphopeptides.

Compared to the unmodified peptides of proteins, phosphopeptides and glycopeptides exist at a substoichiometric level. Phosphorylated peptides with a negative charge are common in positive-ion-mode mass spectrometry because their ionization efficiency is lower than that of non-phosphorylated peptides. It is necessary to apply the corresponding enrichment technology to solve the problems caused by the above problems. Paul J. Boersema [15] focused on the specific fragmentation of phosphopeptides in MS and how such data could be used to identify phosphopeptides and locate phosphorylation sites. The preparation method of the sample, the selection of protease and whether to use additional HPLC separation dimensions should be taken into account. This will be a very large and complex work, but the enrichment of peptides in advance can greatly improve the efficiency of the work.

Based on the retention difference of different phosphopeptides under hydrophilic interaction chromatography (HILIC), Dean E. McNulty [54] uses the strong hydrophilicity of phosphate groups to selectively enrich and separate phosphopeptides. This method not only consumes a lot of time, but also consumes a lot of mobile phases before optimizing the gradient. In addition, the fraction also has continuity, which makes it difficult to completely separate the required phosphopeptides and glycopeptides.

On this basis, we use diaminoferrocene and dialdehyde-based ferrocene as raw materials to prepare ferrocene-based COF (FE-1 COF) materials through a reversible imine condensation reaction. It has the characteristics of imide COF materials, including a large specific surface area, high porosity, a controllable structure and good hydrophilicity [55,56,57]. Meanwhile, the presence of Fe metal ions in ferrocene-based COF has potential applications in the enrichment of phosphopeptides. Therefore, the material has a high potential for the enrichment of glycopeptides and phosphopeptides. By optimizing the loading conditions and elution conditions, the purpose of selectively enriching phosphopeptides and glycopeptides can be achieved.

## 2. Materials and Methods

### 2.1. Material and Reagents

1,1′-Ferrocene dicarbaldehyde and 1,1′-Ferrocene dimethanol were both analytical grade and were purchased from Yanfeng Technology (Shenyang, China). Glacial acetic acid, dichloromethane, ethanol, ethyl acetate, acetone, mesitylene, 1,4-dioxane and N,N-dimethylformamide (DMF) are all analytically pure; concentrated ammonia solution (NH_3_·H_2_O, 25 wt%), acetonitrile (ACN, HPLC) and trifluoroacetic acid (TFA, 99%) were purchased from Aladdin (Shanghai, China). Trypsin (TPCK-treated trypsin, 10,000 U mg^−1^), α-casein (α-casein), β-casein (β-casein), bovine serum albumin (BSA, 98%), 2,5-Dihydroxybenzoic acid (DHB, 99%), dithiothreitol (DTT, 99%), PN-Gase F (glycerol-free, recombinant, 15,000 U), iodoacetamide (IAA, 99%), Immunoglobulin G (IgG), urea and ammonium bicarbonate (NH_4_HCO_3_) were purchased from Sigma Aldrich (St Louis, MO, USA). Healthy human serum was provided by East China University of Science and Technology Hospital.

### 2.2. Synthesis of FE-1 COF Material

Synthesis of monomer diaminoferrocene:(1)Weigh 200 mg of 1,1′-ferrocene dimethanol, dissolve it in glacial acetic acid (8 mL), add 520 mg of sodium azide and stir for 4–6 h at 50 °C. Rotate in vacuo to remove the solvent. Partition the residue between dichloromethane (3 × 15 mL) and saturated sodium bicarbonate solution, and the organic salt layer was washed with saturated brine. The material is dried and filtered with anhydrous sodium sulfate.(2)The obtained 1,1′-bis(azidomethyl)ferrocene oily liquid (174 mg) was dissolved in ethanol (4.8 mL) and water (1 mL), and 143 mg NH_4_Cl and 149 mg were added to activated zinc powder. The suspension was stirred vigorously at room temperature until TLC analysis showed complete conversion of the substrate. After filtration, the filtrate was partitioned between ethyl acetate (3 × 20 mL) and 1 mol/L ammonia solution (14 mL). The organic layer was separated and washed with saturated brine. Anhydrous sodium sulfate treatment was used for dehydration. The organic solvent was removed by rotary evaporation to obtain the final product. Diaminoferrocene is a deep yellow oily liquid.(3)A Pyrex tube (10 cm) was used as the reaction vessel; diamino ferrocene (0.1 mmol) and dialdehyde ferrocene (0.1 mmol) were dissolved in 3 mL DMAc, 1.5 mL 1,4-dioxane and 0.5 mL acetic acid (6 mol/L). The obtained material is ultrasounded for 5 min and thawed with a freezer pump. The tube was sealed after 3 cycles and left to react at 120 °C for 3d. After the reaction was completed, acetone/DMAC was washed 3 times each. Dry under vacuum at 180 °C overnight. The preparation of ferrocene-based covalent organic material FE-1 is shown in Figure 1.

The final FE-1 COF material had a mass of 40.6 mg, and its purity has been identified as more than 80% by melting point method.

### 2.3. Preparation of Tryptic Digests of Standard Proteins and Biological Samples

Preparation of standard phosphoprotease hydrolysis solution: Weigh 1 mg of standard phosphoprotein α-casein, and dissolve it in 1 mL of 50 mmol/L ammonium bicarbonate (pH = 8.3) solution, according to the ratio of enzyme to protein of 1:25. Add trypsin (0.025 mg) at a ratio, and carry out enzymatic hydrolysis at 37 °C for 17 h. Weigh 2 mg bovine serum albumin and dissolve it in 1 mL solution containing 8 mol/L urea and 50 mmol/L ammonium bicarbonate (pH = 8.3), and add 20 μL 1 mol/L dithiothreitol at 60 °C. Incubate for 1 h and add 7.2 mg of iodoacetamide, in a dark room, for the alkylation reaction for 45 min, with 50 mmol/L ammonium bicarbonate (pH = 8.3) solution diluted 10 times, according to the ratio of enzyme to protein mass. Add trypsin at a ratio of 1:25 and enzymatically digest at 37 °C for 17 h. After the enzymolysis is completed, place lyophilize in aliquots and store in a refrigerator at −20 °C for later use.

Human serum pretreatment: 22.5 μL of loading solution 1 (ACN-H_2_O-TFA, 50/49/1, *v*/*v*/*v*) was added to 2.5 μL of human serum, and the mixture was incubated at 25 °C for 30 min. Then, the above mixture was centrifuged at 1000 rmp for 15 min at room temperature, and take the supernatant for use. Next, 30 μL of skimmed milk and 25 mmol/L ammonium bicarbonate solution were mixed well. The mixture was centrifuged at 15,000 rmp for 15 min, the supernatant was taken, 40 μg trypsin was added and the mixture was reacted at 37 °C for 16 h. After the reaction is over, store in a refrigerator at −20 °C for later use.

### 2.4. Enrichment of N-Glycopeptides and Phosphopeptides

#### 2.4.1. The Process of Glycopeptide Enrichment

Separation and enrichment of glycopeptides in standard protein IgG enzymatic hydrolysis solution: using sample solution (5% TFA, 95% ACN-H_2_O), dilute the standard proteolysis solution to the required concentration, add 50 μg FE-1 COF to 200 mL of the loading solution, incubate for 30 min, centrifuge, discard the supernatant and then wash with the loading solution 3 times. Remove non-specifically adsorbed non-glycopeptides, centrifuge, discard the cleaning solution and finally, add 10 μL of eluent (5% TFA, 30% ACN-H_2_O) for elution, elute twice and use MALDI-TOF as the eluent MS analysis. The composition of the loading solution is 5% TFA, 85% ACN-H_2_O and the other separation and enrichment processes are the same. The separation and enrichment process is shown in Figure 2.

#### 2.4.2. The Process of Phosphopeptide Enrichment

Enriching phosphopeptides in the standard proteolysis solution: 10 µg particles are added to 200 µL loading solution (50% ACN/1% TFA), and 15 pmol α-casein is added for hydrolysis. Incubate by shaking at room temperature for 30 min. For magnetic separation, discard the supernatant, and wash 3 times with 200 μL loading solution (50% ACN/1% TFA). For magnetic separation, discard the supernatant, add 10 μL NH_3_·H_2_O (10 wt%) for elution, eluate twice and analyze the eluate by MALDI-TOF MS.

### 2.5. Mixing Enrichment of Glycopeptides and Phosphopeptides

Next, 10 µg of particles is added to 200 μL of loading buffer (ACN/H_2_O/TFA = 90:8:2, *v*/*v*/*v*) Dilute. After incubating for 30 min at room temperature, the nanocomposite was collected with a magnet, and then washed with 200 μL of buffer (ACN/H_2_O/TFA = 90:9.9:0.1, *v*/*v*/*v*) 3 times. Finally, the captured phosphopeptides and glycopeptides were eluted with different elution buffers (10 μL each of NH_3_·H_2_O and ACN/H_2_O/TFA, 30/65/5, *v*/*v*/*v*) and analyzed by MALDI-TOF MS.

### 2.6. Separation and Enrichment of Glycopeptides in Human Serum

Pipette 2 μL of human serum, add 18 μL of 8 mol/L urea in 50 mmol/L NH_4_HCO_3_ solution, continue to add 1 μL of 200 mmol/L DTT, place in the water bath at 56 °C for 1 h and then, add 4 μL of 200 mmol/L DTT /L of IAA and place in the dark for 45 min. Subsequently, add 5 μL of 1 mg/mL trypsin solution and 50 μL of 50 mmol/L NH_4_HCO_3_ solution. Finally, in a water bath at 37 °C for 17 h, the enzymatic lysate was lyophilized and placed in a −20 °C refrigerator for later use.

Taking 2 μL of the enzymatic hydrolysis solution of the human serum treated above, add 400 μL of loading solution (4% TFA, 90% ACN-H_2_O) to dilute, and then add 50 μg of particles at room temperature. Incubate with shaking for 30 min. Centrifuge, discard the supernatant and wash 3 times with 400 μL of loading solution. Centrifuge, discard the supernatant and add 100 μL of eluate (4% TFA, 30% ACN-H_2_O) for elution. After carrying out elution 3 times, the eluate is combined, lyophilized and re-dissolved in 17 μL of H_2_O, in a mixture of 2 μL glycan buffer and 1 μL PN-Gase F (500U). Incubate at 37 °C for 17 h to remove glycans. The desugared samples were analyzed by nano LC-MS/MS.

### 2.7. Mass Spectrometry Analysis and Data Analysis

All data of MALDI-TOF MS for glycopeptides and phosphopeptides were obtained from an AB Sciex 4800 Plus MALDI TOF/TOF analyzer (AB Sciex, Boston, Ma, USA). Set the wavelength to 335 nm in reflector positive mode with Nd/YAG laser. The accelerating voltage was set to 20 kV, and the mass-to-charge ratio scan range was 1000–3500. A matrix of aqueous acetonitrile (70%, *v*/*v*) containing DHB (25 mg/mL) and H_3_PO_4_ (1%, *v*/*v*) should be prepared in advance. Drop 0.5 μL of eluate onto a MALDI plate and after evaporation, add 0.5 μL of matrix for MS analysis.

All database searches use Proteome Discoverer software (Thermo Fisher Scientific, version 1.4, Waltham, Ma, USA) and the Sequest HT search engine; the database is Human UniProtKB/SwissProt database (Release 2018 01 26, sequence 20189). The data retrieval parameters are as follows: precursor mass tolerance 10 ppm and fragment mass tolerance 0.05 Da. Trypsin digestion can occur twice without lysis. At the peptide and protein level, the false discovery rate is controlled below 1% through the percolator algorithm. For glycopeptides, the aminomethyl group on cysteine is set as a fixed modification. The oxidation of methionine and the deamidation of asparagine are used as variable modifications. Only glycopeptides with N-glycosylation consensus sequence are considered reliable. For phosphopeptides, the aminomethyl group on cysteine is set as a fixed modification.

## 3. Results and Discussion

### 3.1. Synthesis and Characterization 

FE-1 COF is synthesized by the solvothermal method. The morphology of FE-1 COF was observed by SEM and TEM. As shown in Figure 3a, FE-1 COF is a dispersed nano-polymer material. After a longer reaction time, the COF crystallites aggregate to form spherical particles with a smooth surface (3–5 µm, Figure 3).

The element composition of FE-1 COF was determined by XPS spectroscopy. Fe, C, O and N are the four main components of FE-1 COF (Figure 4). Additionally, 708.3 Ev and 721.3 Ev are the binding energy peaks of Fe element 2p 3/2 and 2p 1/2 orbitals, respectively, which are consistent with the XPS test results of other ferrocene-based materials.

The bond formation of FE-1 COF was studied from the FT-IR spectrum data. As shown in Figure 5, a -C=N peak appears at 1647 cm^−1^. Furthermore, 550 cm^−1^ is the characteristic absorption peak of ferrocene. The IR spectrum showed that FE-1 COF was successfully prepared.

FE-1 COF has weak magnetic properties, and the saturation magnetization (Ms) of FE-1 COF is 3.6 emu/g (Figure 6).

The N_2_ adsorption–desorption curve is measured at 77 K, and the specific surface area and pore size of the material are calculated. As shown in Figure 7, after calculation, the specific surface area of FE-1 COF is 61.75 m^2^·g^−1^ and the pore diameter is 1.26 nm. The microporous structure of the covalent organic layer can exclude relatively large-sized proteins during phosphopeptide enrichment [58].

### 3.2. Selective Enrichment of Phosphopeptides and Determination of Enrichment Capacity

The presence of a large amount of metallic Fe ions in the FE-1 COF material gives the material a good enrichment effect for phosphopeptides. Tryptic hydrolysates of α-casein and β-casein were selected as model samples to evaluate the phosphopeptide enrichment performance of the synthesized materials. After the separation and enrichment process of sample loading, washing and elution, it was detected by MALDI-TOF MS. As shown, direct mass spectrometry analysis of 2 μg of α-casein and β-casein tryptic digests detected only three low-intensity phosphopeptides, respectively (Figure 8). Due to the influence of non-phosphorylated peptides and salts in the enzymatic hydrolysate, the peaks in the mass spectrum are mainly the mass spectrum peaks of high-abundance non-phosphorylated peptides, and the mass spectrum signal intensity of phosphopeptides is low. After enrichment of FE-1 COF material, 18 and 8 phosphopeptides were captured, respectively, and the signal intensity had been greatly improved (Figure 9), indicating its selective enrichment ability for phosphopeptides. Detailed information on phosphopeptides detected from α-casein and β-casein tryptic digest is listed in Appendix A.

Firstly, the adsorption properties of the materials were evaluated. Phosphopeptides were enriched from 2 µg α-casein tryptic digests by materials of different qualities. Among the detected phosphopeptides, three phosphopeptides (m/z = 1660, 1927 and 1951) were selected as detection markers. As shown in Figure 10, phosphopeptides are enriched by increasing the amount of material. When the amount of material reaches 5 µg, the signal intensity of the three phosphopeptides reaches the maximum. The phosphopeptide enrichment capacity was calculated to be 300 mg/g (Figure 10).

### 3.3. Selective Enrichment of Glycopeptides

The standard glycoprotein IgG is selected as the glycopeptide-enriched sample. For the separation and enrichment of glycopeptides, the selection of enrichment conditions is very important. In order to obtain the best enrichment effect, we adjusted the TFA concentration in the loading solution. As shown in Figure 10, the TFA concentration in the loading solution gradually increased from 0.1% to 6%. When the TFA concentration is 0.1–2%, after FE-1 enrichment, no one glycopeptide can be observed from the mass spectrum. When the TFA concentration is 3%, only a few glycopeptides can be observed. When TFA is 4% to 6%, the effect of enriching glycopeptides is good. Therefore, considering the signal intensity and the number of bars observed from the mass spectrum, 4% is selected as the optimal TFA concentration for loading. The elution buffer is still 0.4 mol·L^−1^ NH_3_·H_2_O. Under the optimal conditions, after FE-1 enrichment, 21 glycopeptides were detected from the IgG hydrolysate (Figure 11), which shows that FE-1 can selectively enrich glycopeptides. Detailed information on glycopeptides detected from IgG tryptic digest is listed in Appendix A.

### 3.4. Simultaneous Enrichment of Phosphopeptides and Glycopeptides

Considering that both phosphopeptides and glycopeptides exist in biological samples, FE-1 COF can be further applied to the simultaneous enrichment of phosphopeptides and glycopeptides. Standard IgG and α-casein were selected as model samples because their tryptic hydrolysates contain a large amount of glycopeptides and phosphopeptides, which makes it possible to enrich simultaneously in biological samples.

The enrichment and elution mechanisms of phosphopeptides and glycopeptides are different, and the concentration of ACN and TFA in the loading buffer will affect the enrichment efficiency of glycopeptides and phosphopeptides, so separate enrichment and stepwise elution methods are used. In this way, simultaneous enrichment of phosphopeptides and glycopeptides in IgG and α-casein tryptic hydrolysates.

The concentration of ACN in the loading buffer was first adjusted. Different concentrations of ACN containing 0.1% TFA were tested, and the captured glycopeptides after enrichment were first washed with elution buffer 1 (3 × 10 μL, ACN/H_2_O/TFA, 30:65:5, *v*/*v*/*v*) After elution, the captured phosphopeptides were then eluted with elution buffer 2 (3 × 10 μL, NH_3_·H_2_O, 0.4 mol/L).

As shown in Figure 12, the concentration of ACN has a certain effect on the enrichment of phosphopeptides, but a greater impact on the enrichment of glycopeptides. Therefore, ACN with a concentration of 90% was selected. The reason is that glycopeptides cannot be completely retained by the material at a relatively low ACN concentration; when the ACN concentration is too high, glycopeptides cannot be completely dissolved in the loading buffer, resulting in a decrease in enrichment performance.

Next, different concentrations of TFA containing 90% ACN were tested, and the elution conditions were the same as above. As shown in Figure 13, the number of phosphopeptides and glycopeptides detected varied. When the concentration of TFA was 1%, the number of enriched glycopeptides and phosphopeptides was the highest. Because TFA is an ion-pairing reagent that can label the difference in hydrophilicity between glycopeptides and non-glycopeptides in the loading buffer, the addition of TFA can improve the enrichment efficiency of peptides. On the other side, excess TFA leads to a decrease in the number of enriched phosphopeptides.

During elution, the phosphopeptide is eluted after three elutions of the glycopeptide, which may lead to the loss of the phosphopeptide. In consideration of the enrichment efficiency of phosphopeptides and glycopeptides, the loading solution of ACN/H_2_O/TFA (90/9/1, *v*/*v*/*v*) and the elution buffer 1 of glycopeptides of ACN/H_2_O/TFA were selected (30/65/5, *v*/*v*/*v*) as optimal conditions. As shown in Figure 14, ACN/H_2_O/TFA (90/9/1, *v*/*v*/*v*) was used as the sample solution, followed by eluent 1 (ACN/H_2_O/TFA, 30/65/5, *v*/*v*/*v*) and elution buffer 2 (NH_3_·H_2_O, 0.4 mol/L); 19 glycopeptides and 16 phosphopeptides were simultaneously captured. Detailed information of glycopeptides detected from IgG tryptic digest and phosphopeptides detected from α-casein tryptic digest is listed in Appendix A.

In addition, the stability and repeatability of using the FE-1 COF material for the simultaneous enrichment of phosphopeptides and glycopeptides were investigated in α-casein and IgG tryptic digest. On the one hand, stability was assessed by storing the material for 6 months at normal temperature. The results indicated that the numbers of captured phosphopeptides and glycopeptides were considerable, and the signal intensity of them still remained relatively high. As for repeatability, the material should be washed with an elution buffer to remove residual peptides before the next cycle. The number of captured phosphopeptides and glycopeptides and intensity basically remained unchanged after three cycles. All the data show that the FE-1 COF material has great stability at a normal temperature and repeatability for multiple enrichments (Figure 15).

### 3.5. Selective Enrichment of Phosphopeptides and Glycopeptides in Biological Samples

For the enrichment of phosphopeptides and glycopeptides, human serum is a common biological sample, which contains four characteristic endogenous phosphopeptides and a large number of glycoproteins with a complex background. Therefore, it can be used to evaluate the enrichment effect of this material on low-abundance endogenous phosphopeptides and glycopeptides in complex backgrounds. As a result, 4 endogenous phosphopeptides (Figure 16) and 41 N-glycopeptides were identified from 4 μL of human serum trypsin digest after enrichment of FE-1 COF material.

## 4. Conclusions

In this paper, based on the principle of the imine condensation reaction, a new ferrocene-based covalent organic material FE-1 was prepared by the traditional solvothermal method. This material is micron-sized spherical particles, in which there are a large number of amino groups and Fe metal ions which can be used for the enrichment of phosphopeptides and glycopeptides. Compared with the interference of other peptides by direct MS analysis, the enrichment of phosphopeptides and glycopeptides through FE-1 COF material synthesized by us in advance can avoid the impact of sample complexity. After further optimization of conditions, the gradual elution and simultaneous enrichment of phosphopeptides and glycopeptides are realized. A good enrichment effect has been achieved. On the other hand, it also has a certain phosphopeptides and glycopeptides enrichment effect in complex biological samples, demonstrating its great potential in proteomics analysis.

## Figures and Tables

**Figure 1 jfb-15-00185-f001:**
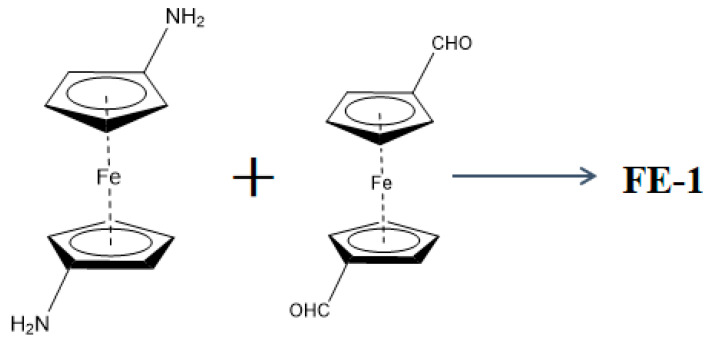
Preparation diagram of ferrocene covalent organic material FE-1.

**Figure 2 jfb-15-00185-f002:**
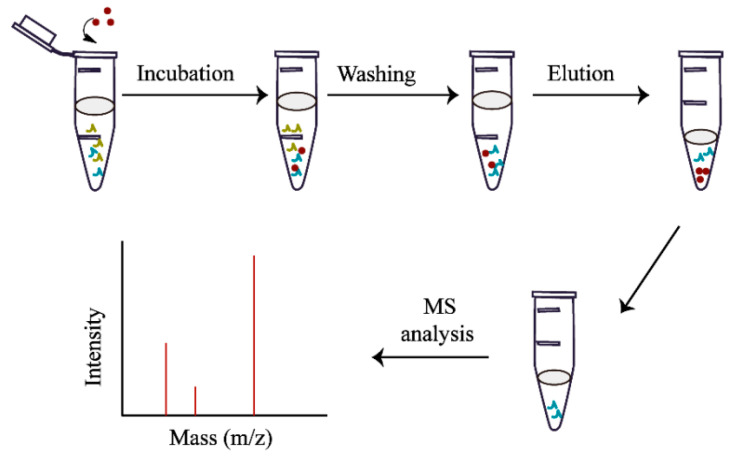
Flow chart for isolation and enrichment of glycopeptides.

**Figure 3 jfb-15-00185-f003:**
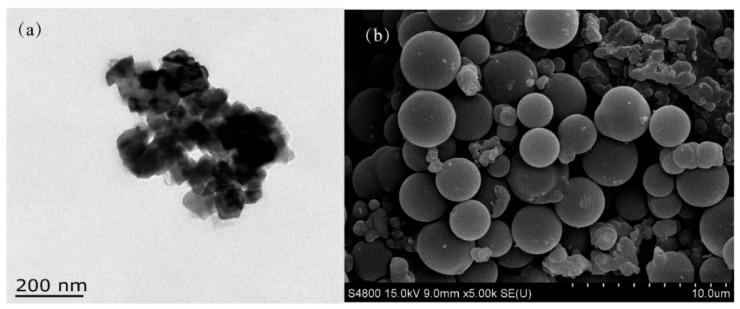
(**a**) TEM image, (**b**) FE-SEM image of FE-1.

**Figure 4 jfb-15-00185-f004:**
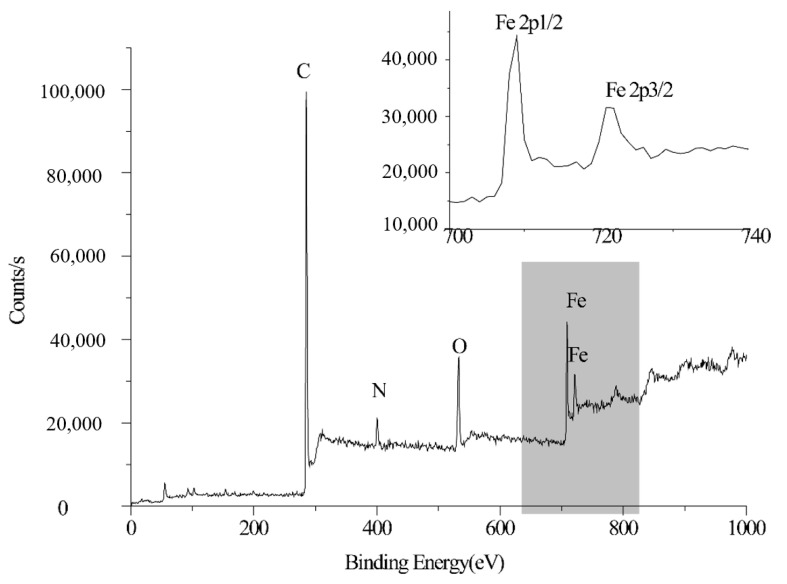
XPS spectra of FE-1.

**Figure 5 jfb-15-00185-f005:**
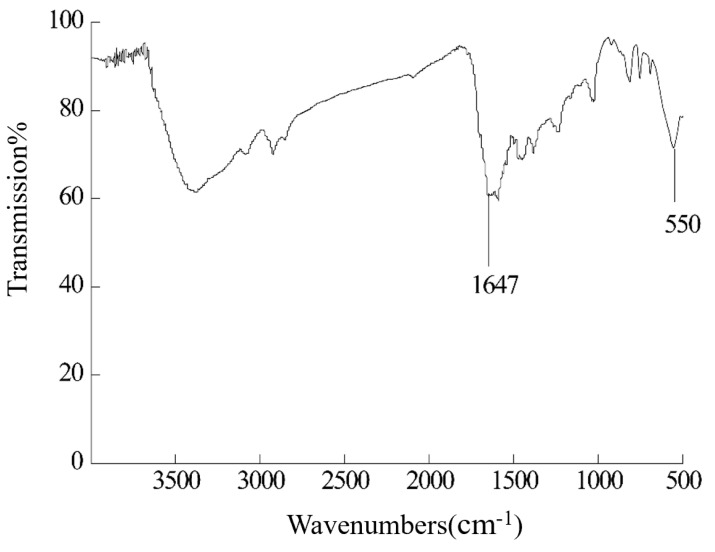
IR spectra of FE-1.

**Figure 6 jfb-15-00185-f006:**
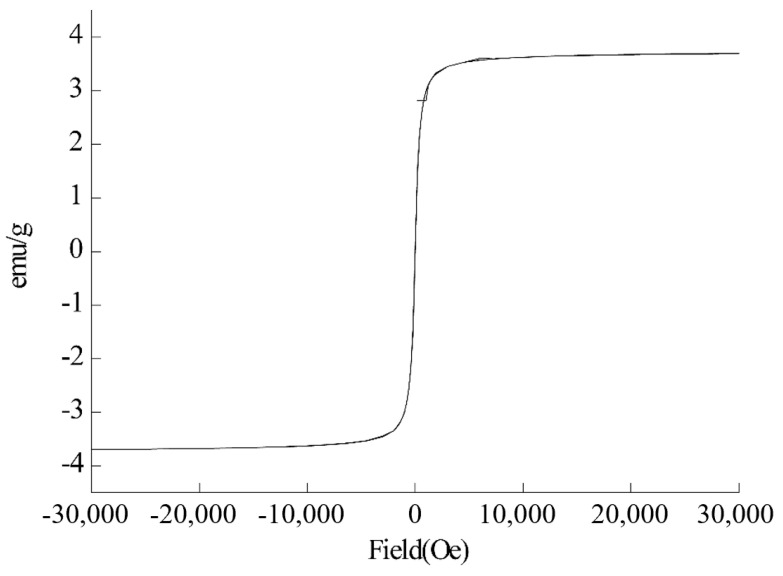
Magnetic hysteresis curve.

**Figure 7 jfb-15-00185-f007:**
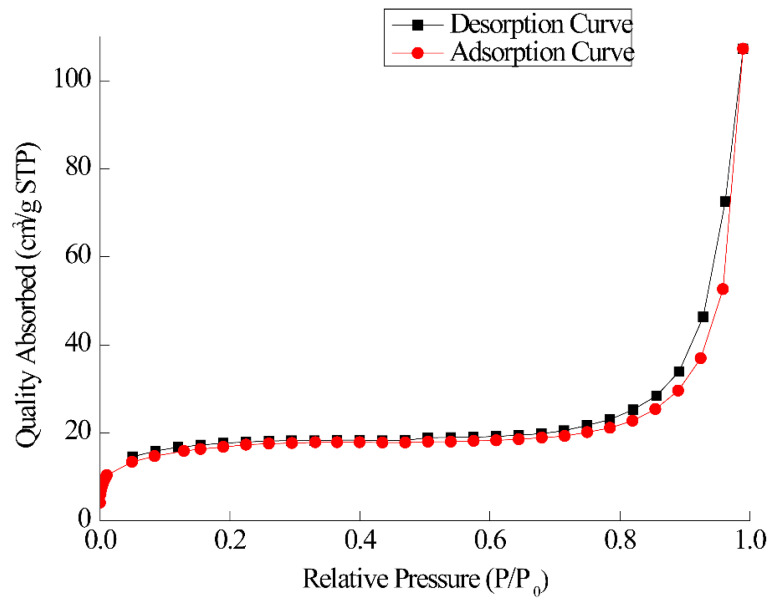
N_2_ adsorption–desorption curve of FE-1.

**Figure 8 jfb-15-00185-f008:**
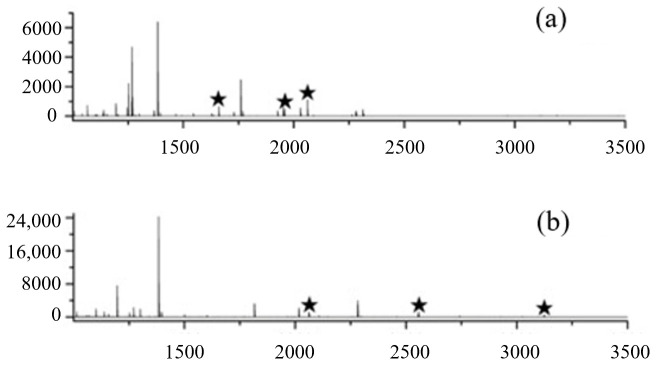
(**a**) MALDI-TOF mass spectra of the tryptic digests of α-casein. (**b**) MALDI-TOF mass spectra of the tryptic digests of β-casein.

**Figure 9 jfb-15-00185-f009:**
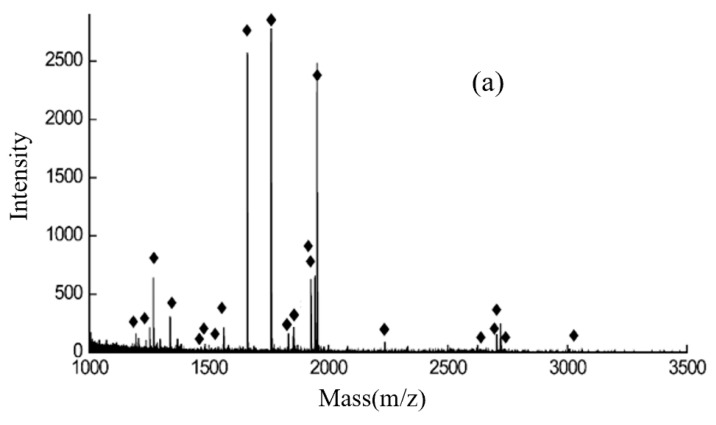
(**a**) Mass spectrum of the eluate after enrichment with FE-1 COF material (α-casein). (**b**) Mass spectrum of the eluate after enrichment with FE-1 COF material (β-casein).

**Figure 10 jfb-15-00185-f010:**
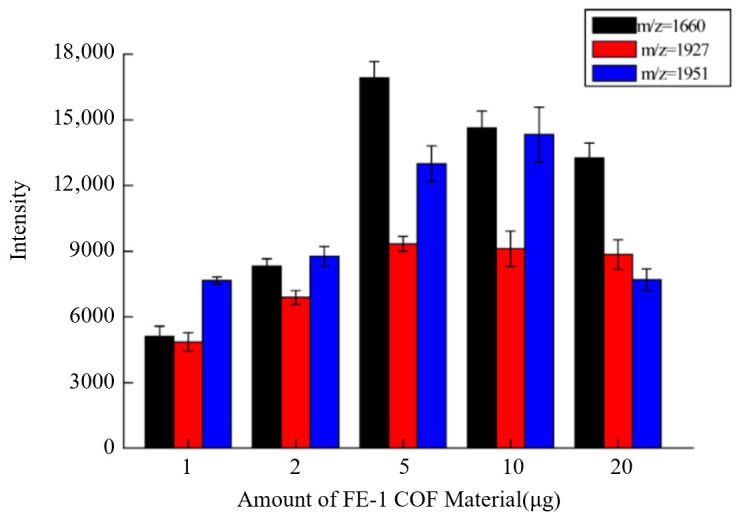
Detection of phosphopeptides enrichment capacity of FE-1 COF material from 2 µg α casein tryptic digest.

**Figure 11 jfb-15-00185-f011:**
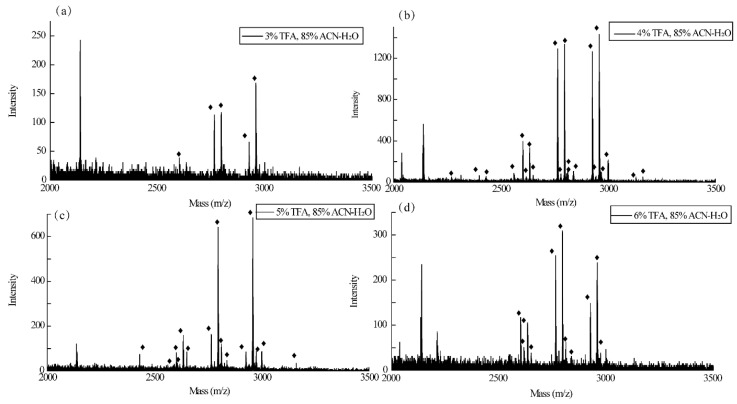
MALDI-TOF mass spectra of the tryptic digest of 2 μL 1 mg/mL IgG enriched by 100 μg FE-1 COF in different acidity of loading buffer.(**a**) 3% TFA; (**b**) 4%TFA; (**c**) 5% TFA; (**d**) 6% TFA.

**Figure 12 jfb-15-00185-f012:**
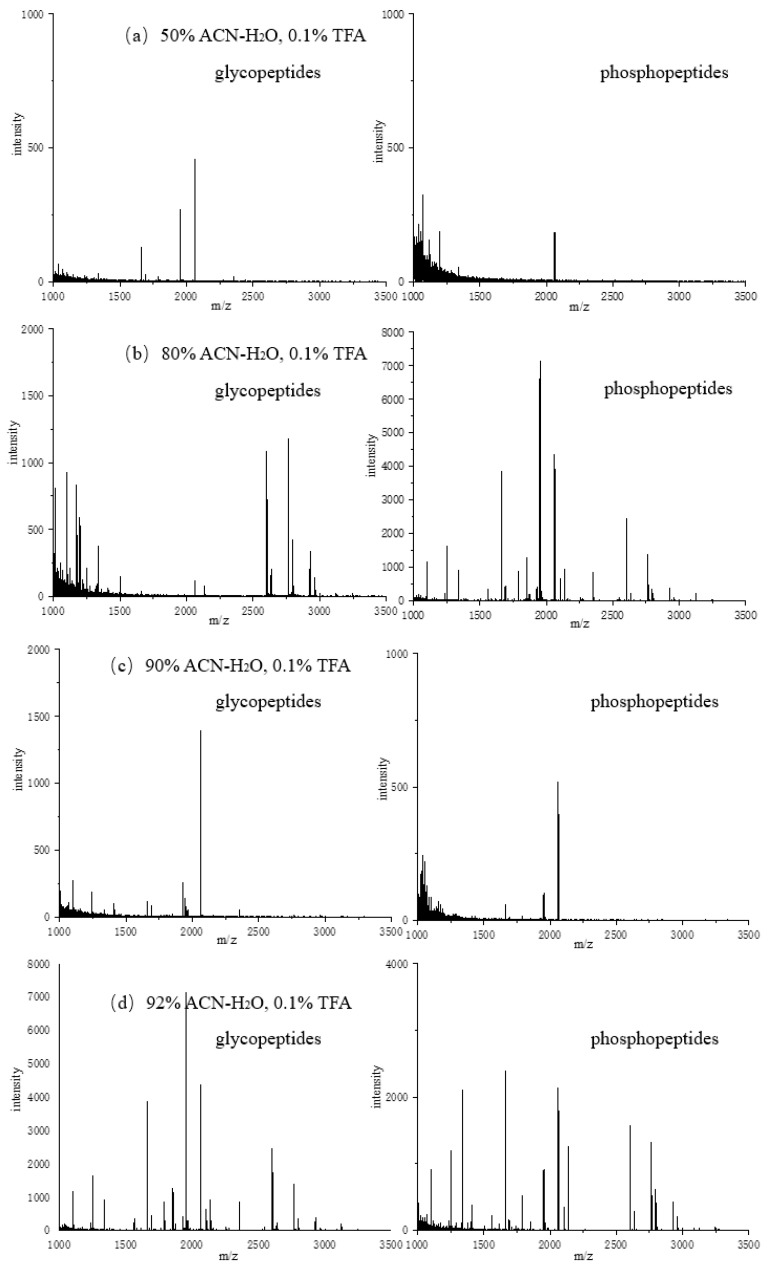
Mass spectra of glycopeptides and phosphorylated peptides enriched in loading buffers containing different concentrations of ACN: (**a**) 50% ACN; (**b**) 80% ACN; (**c**) 90% ACN; (**d**) 92% ACN.

**Figure 13 jfb-15-00185-f013:**
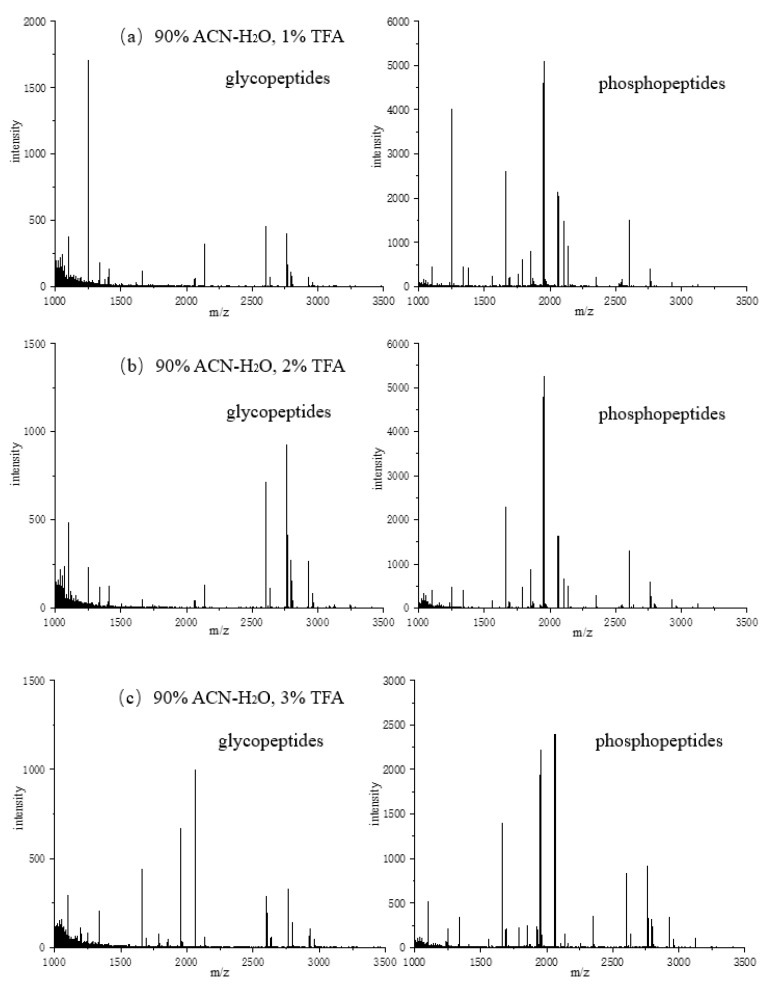
Mass spectra of glycopeptides and phosphorylated peptides in loading buffer enriched with different concentrations of TFA: (**a**) 1% TFA; (**b**) 2% TFA; (**c**) 3% TFA; (**d**) 4% TFA; (**e**) 5% TFA.

**Figure 14 jfb-15-00185-f014:**
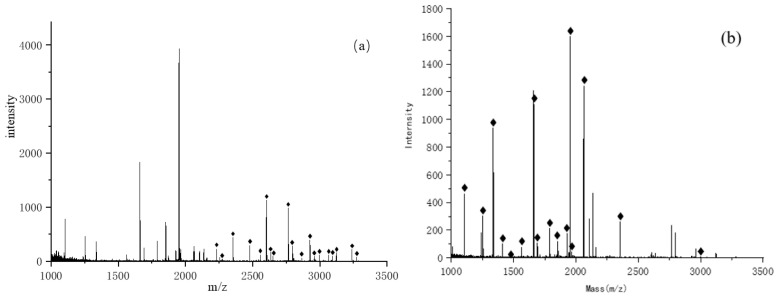
(**a**) Mass spectrum of eluent 1 obtained by enriching glycopeptides in IgG enzymatic hydrolysate with FE-1 COF material; (**b**) Mass spectrum of eluent 2 obtained by enriching phosphopeptides in α-casein enzymatic hydrolysate with FE-1 COF material.

**Figure 15 jfb-15-00185-f015:**
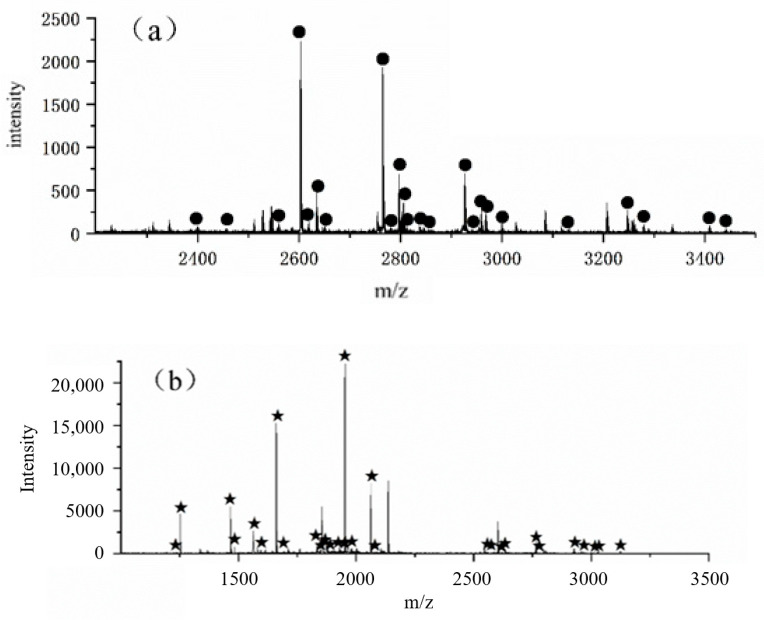
After storing for 6 months and three cycles, (**a**) Mass spectrum of eluent 1 obtained by enriching glycopeptides in IgG enzymatic hydrolysate with FE-1 COF material; (**b**) Mass spectrum of eluent 2 obtained by enriching phosphopeptides in α-casein enzymatic hydrolysate with FE-1 COF material.

**Figure 16 jfb-15-00185-f016:**
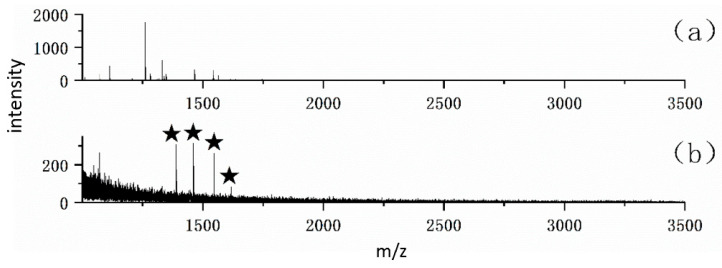
MALDI-TOF mass spectra of 4 μL of human serum trypsin digest (**a**) before, (**b**) after enrichment by 100 μg FE-1 COF material.

## Data Availability

The data presented in this study are available on request from the corresponding authors.

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
