# Peer review of "A Type of Ferrocene-Based Derivative FE-1 COF Material for Glycopeptide and Phosphopeptide Selective Enrichment"

_jfb, 2024, doi:10.3390/jfb15070185_

Round 1

Reviewer 1 Report

Comments and Suggestions for Authors

The manuscript by Yu Wu et al presents an approach towards the development of a ferrocene-based covalent organic framework (FE-1 COF) material aimed at the selective enrichment of glycopeptides and phosphopeptides. This work is relevant to the field of proteomics, as it addresses the challenge of analyzing post-translational modifications (PTMs), which are pivotal in understanding biological processes and disease mechanisms. Through comprehensive synthesis, characterization, and application methodologies, the authors convincingly demonstrate the potential of FE-1 COF in enhancing the detection of PTMs from complex biological samples. The utilization of various characterization techniques to elucidate the properties of FE-1 COF, alongside experimental data, is well acceptable. 

1)       Despite the manuscript's strengths, I have identified areas that would benefit from further clarification to bolster the study's contributions to the field.

2)       The manuscript briefly mentions existing materials for phosphopeptide and glycopeptide enrichment but lacks a direct comparative analysis. How does FE-1 COF perform relative to other state-of-the-art materials, especially in terms of selectivity, capacity, and compatibility with mass spectrometry?

3)       Could you provide more details on the yield and purity of the FE-1 COF material post-synthesis? Should be discussed in the main manuscript.

4)       The practical application of FE-1 COF in proteomics would greatly benefit from data on its stability over time and its reusability. Are there any studies on the stability of FE-1 COF under storage conditions and its performance after multiple enrichment cycles?

Overall, the manuscript is well-written, with appropriate and relevant references cited, and the data are well presented with relevant data. The study makes a significant contribution to the advancement of biomaterials for proteomics research. Therefore, I recommend this manuscript for publication in the MDPI's "Journal of Functional Biomaterials" after the authors address the aforementioned technical questions.

Comments on the Quality of English Language

Need to be checked typo in whole manuscript. 

Author Response

Thank you very much for your modification suggestions. I have revised the manuscript according to your suggestion, please refer to the attachment for the specific reply.

Reviewer 2 Report

Comments and Suggestions for Authors

The search for new materials for the selective enrichment of modified peptides, occurring in trace amounts, is extremely important and undertaken by scientists.

The manuscript should be revised according to the following question and comments:

1. Have any studies been conducted on the relationship between the amount of FE-1 COF material and the amount of material that is enriched?

2. Were unmodified peptides enriched on the FE-1 COF material in addition to phosphopeptides and glycopeptides? Please post original data from your peptide identification program to show this.

3. What was the sequence coverage of analyzed proteins based on the identified, modified peptides using the mentioned database?

4. The MALDI-MS spectra presented in this form without signal signatures provide very little information. Some of the data can be transferred to SI and the optimization of the elution from the material can be considered and presented in the form of a graph.

5. The presented tables of identified peptides should include comments and comparisons with other previously used enrichment methods. I mean conducting experiments on your own, using other enrichment methods, and comparing them with the proposed new material. Did this enrichment method identify more peptides? And whether a sample was tested, e.g. LC-MS/MS without enrichment, to show that all modified fragments were identified and that the enrichment method works. And that no modified peptides are lost. Without showing a comparison, we don't know if it works better or just works. Without this information, I do not support the acceptance of this manuscript.

6. In addition to showing the results of the experiment, please show that the new enrichment method is better than the previous ones because this is definitely not evident from the conclusions contained in the paper.

Author Response

(The authors gave the same response as above.)

Round 2

Reviewer 2 Report

Comments and Suggestions for Authors

Regarding comment 6. In typical proteomic studies, to show that the proposed method is better, it needs to be compared with previous methods. This has not been included in the conclusions, although it is mentioned that both phosphorylated and glycosylated peptides can be enriched, which is undoubtedly an advantage. In typical proteomic studies, to verify the developed enrichment method, an LC-MS/MS analysis of the hydrolysate is performed before and after using the enrichment materials, and then the obtained results allow us to determine the number of modified peptides identified and conclude. In the case of the Maldi method, unfortunately, we do not have LC separation, so direct analysis before enrichment will certainly not be a good solution. I missed this information as a summary, pointing out the advantages of the MALDI method combined with enrichment. Please complete this. I appreciate the quality of the work that has been done.

Author Response

Regarding comment 6. In typical proteomic studies, to show that the proposed method is better, it needs to be compared with previous methods. This has not been included in the conclusions, although it is mentioned that both phosphorylated and glycosylated peptides can be enriched, which is undoubtedly an advantage. In typical proteomic studies, to verify the developed enrichment method, an LC-MS/MS analysis of the hydrolysate is performed before and after using the enrichment materials, and then the obtained results allow us to determine the number of modified peptides identified and conclude. In the case of the MALDI method, unfortunately, we do not have LC separation, so direct analysis before enrichment will certainly not be a good solution. I missed this information as a summary, pointing out the advantages of the MALDI method combined with enrichment. Please complete this. I appreciate the quality of the work that has been done.

Response:

       I have added a discussion on the direct use of MS analysis at the end of Introduction and pointed out the advantages of combining MALDI and enrichment in the conclusion.
       Thank you very much for your valuable advice. Please let me know if there is any data that needs to be added
